# Somatostatin analog therapy effectiveness on the progression of polycystic kidney and liver disease: A systematic review and meta-analysis of randomized clinical trials

**Tatsuya Suwabe**[1,2]*, **Francisco J. Barrera**[3,4], **Rene Rodriguez-Gutierrez**[3,4], **Yoshifumi Ubara**[1,2], **Marie C. Hogan**[5]

**1** Division of Nephrology, Toranomon Hospital, Tokyo, Japan, **2** Okinaka Memorial Institute for Medical Research, Toranomon Hospital, Tokyo, Japan, **3** Plataforma INVEST Medicina UANL KER Unit Mayo Clinic, KER Unit México, Medical School and University Hospital "Dr. Jose E. Gonzalez", Universidad Autonoma de Nuevo Leon, Monterrey, Mexico, **4** Knowledge and Evaluation Research Unit in Endocrinology (KER-Endo), Mayo Clinic, Rochester, MN, United States of America, **5** Division of Nephrology and Hypertension, Mayo Clinic, Rochester, MN, United States of America

* suwabetat@gmail.com

## Abstract

### Background

Uncertainty underlies the effectiveness of somatostatin analogues for slowing the progression of polycystic kidney or liver disease.

### Methods

Eligible studies included randomized controlled trials (RCTs) evaluating somatostatin analog as therapy for patients with polycystic kidney disease (PKD) or polycystic liver disease (PLD) compared to placebo or standard therapy. Two reviewers independently screened studies identified from databases (MEDLINE, EMBASE, Cochrane Database), clinical trial registries, and references from pertinent articles and clinical practice guidelines. Outcome measurements were changes in total liver volume (TLV), total kidney volume (TKV), and estimated glomerular filtration rate (eGFR).

### Results

Of 264 nonduplicate studies screened, 10 RCTs met the inclusion criteria. The body of evidence provided estimates warranting moderate confidence. Meta-analysis of 7 RCTs including a total of 652 patients showed that somatostatin analogs are associated with a lower %TLV growth rate compared to control (mean difference, -6.37%; 95% CI -7.90 to -4.84, $p<0.00001$), and with a lower %TKV growth rate compared to control (mean difference, -3.66%; 95% CI -5.35 to -1.97, $p<0.0001$). However, it was not associated with a difference in eGFR decline (mean difference, -0.96 mL/min./1.73m$^2$; 95% CI -2.38 to 0.46, $p = 0.19$).

**Data Availability Statement:** All relevant data are within the manuscript and its Supporting Information files.

**Funding:** This work was supported by JSPS KAKENHI in the form of a grant awarded to TS (JP19K17758) and in part by the Ministry of Health, Labour and Welfare of Japan in the form of a Grant-in-Aid for Progressive Renal Disease Research and Okinaka Memorial Institute for Medical Research, Toranomon Hospital in the form of funds awarded to TS.

**Competing interests:** The authors have declared that no competing interests exist.

## Conclusions

Current body of evidence suggests that somatostatin analogs therapy slows the increase rate of TLV and TKV in patients with PKD or PLD compared to control within a 3-year follow-up period. It does not seem to have an effect on the change in eGFR. Somatostatin analogs therapy can be a promising treatment for ADPKD or ADPLD, and we need to continue to research its effectiveness for ADPKD or ADPLD.

## Introduction

Autosomal dominant polycystic kidney disease (ADPKD) is the most frequent inherited kidney disease and is worldwide the fourth leading cause of end-stage renal disease (ESRD) in adults [1, 2]. ADPKD is a multisystemic disorder and patients often present with extrarenal manifestations such as polycystic liver disease (PLD) [3–5]. Nonetheless, PLD can also arise in the absence of polycystic kidneys or in the presence of few renal cysts, which is denominated isolated polycystic liver disease (PCLD) or autosomal dominant polycystic liver disease (ADPLD) [6–12].

Evidence supports that somatostatin may blunt cyst development by acting at multiple levels: inhibition of secretin release by the pancreas [13], inhibition of secretin-induced cAMP generation and fluid secretion in cholangiocytes [14–16], vasopressin-induced cAMP generation and water permeability in collecting ducts by its effects on G protein-coupled receptors (Gi subtype), and suppression of the expression of IGF-1, vascular endothelial growth factor, and other cystogenic growth factors causing downstream signaling of their receptors [17–21]. Therefore, theoretically, somatostatin analogs could provide benefit for both these diseases.

Some randomized clinical trials (RCTs) have evaluated the therapeutic effectiveness of somatostatin analogues in these clinical contexts. Moreover, three previous meta-analysis were done to estimate the effectiveness of this therapeutic option [22–24]. Two of them reported that somatostatin analogs attenuated the total kidney volume (TKV) increase rate and did not altered estimated glomerular filtration rate (eGFR) [23, 24]. However, the body of evidence has continued to grow and this meta-analysis did not analyzed the effect of these drugs on the total liver volume (TLV). Another previous meta-analysis reported that somatostatin analogs attenuated TLV, but it did not show demonstrated an improvement in TKV and eGFR [22]. This last study used absolute volumes of kidney and liver which may lead to an under- or over-estimated effect size due to highly heterogeneous baseline volumes between studies. To try to overcome these limitations, we conducted a systematic review and meta-analysis of RCTs using percentage change instead of absolute volumes to assess effectiveness of somatostatin analogs therapy regarding the progression of polycystic kidney or liver disease.

## Methods

This study was conducted following guidance provided by the Cochrane Handbook for systematic reviews [25]; and it is reported in accordance to the recommendations set by the Preferred Reporting Items for Systematic Reviews and Meta-Analyses (PRISMA) work group (S1 Table, S1 Fig). The protocol of this study has been registered the PROSPERO international registry (CRD42018105336). This study protocol is accessed by the Web address (https://www.crd.york.ac.uk/prospero/display_record.php?RecordID=105336).

## Eligibility criteria

We included RCTs published in peer review journals that compared a somatostatin analog against control in adult patients with PKD or PLD within a follow-up timeframe of at least 6 months. The outcomes of interest TLV, TKV or eGFR of patients with PKD or PLD.

## Data sources and search strategy

A comprehensive literature search strategy, with input from study investigators, was designed and carried out by an expert librarian (P.J.E.) using MEDLINE, EMBASE, The Cochrane Database of Systematic Reviews, and The Cochrane Central Register of Controlled Trials databases. The timeframe was from each database inception to May 15, 2018 with no language restriction. The literature search strategy was updated on November 4, 2020 using the same database and the same method by an experienced librarian (E.G.). The complete search strategy can be found on the supplementary material (S2 Table).

## Study selection

The selection process consisted of a title and abstract screening phase and a full- text screening phase (Fig 1). In both phases, each reference was screened independently by two reviewers using standardized pilot-tested instructions. As part of calibration, eligibility criteria were iterated for clarity and consistency. In the title and abstract screening level, both reviewers must have agreed to exclude an article; conflicts were included. Disagreements at the full-text screening phase were resolved by a consensus between both reviewers. When reviewers couldn't reach consensus, a third reviewer was consulted (YU). List of excluded studies are presented in S3 Table. A total of 10 studies were included in this study (Table 1).

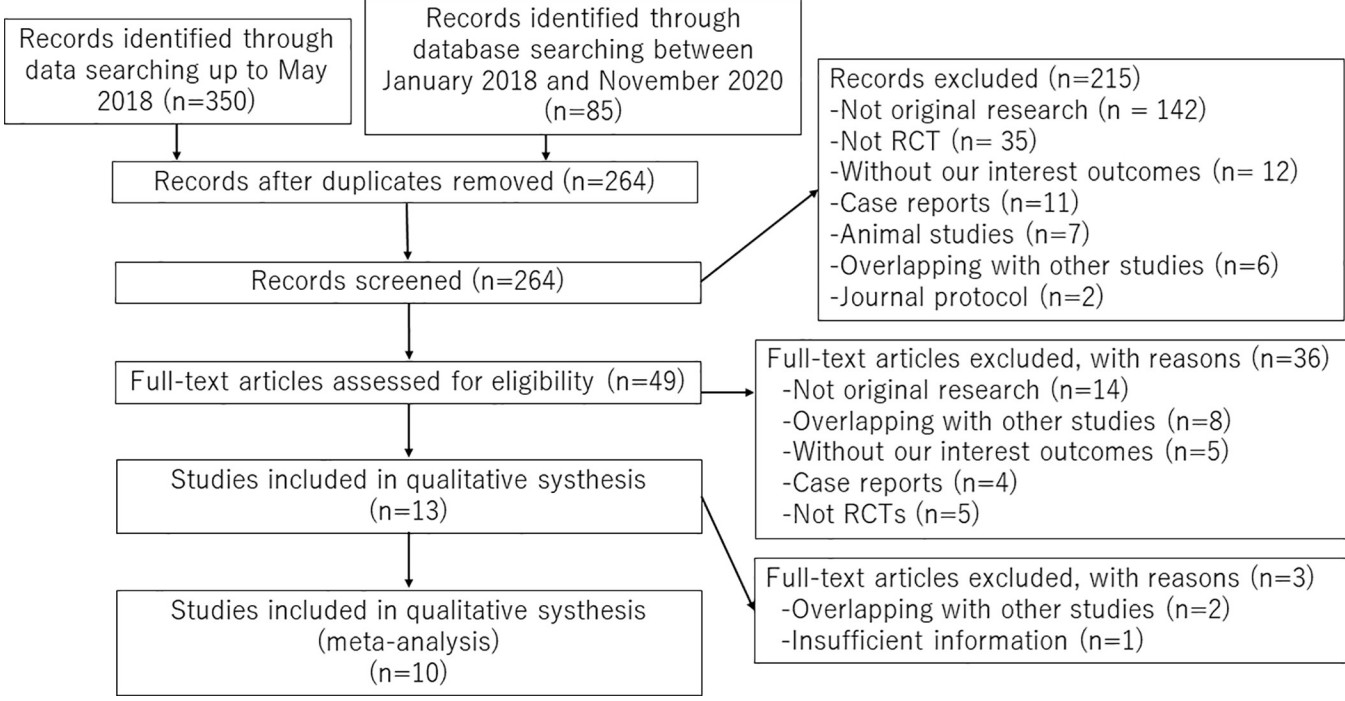

**Fig 1. Process of study selection.**

**Table 1. Characteristics of the included studies.**

| Author, year Study | Country | Study design | N total (% males) | N Somatostatin therapy (% males) | N Control therapy (% males) | Age (mean ± SD) years old | Intervention | Control | Outcome(s) | Follow-up period |
|---|---|---|---|---|---|---|---|---|---|---|
| Ruggenenti et al., 2005 | Italy | Single-center crossover RCT | 12 (75) | 12 (75.0) | 12 (75.0) | 44.5 (35–58) [†] | Octreotide 40mg every 28 days | Placebo | TKV, eGFR | 6 months |
| van Keimpema et al., 2009 (LOCK CYST) | Netherlands and Belgium | Multicenter parallel RCT | 54 (13) | 27 (11.1) | 27 (14.8) | Intervention: 49.6 (34.4–64.8) [‡] <br> Control: 50.3 (32.6–68.1) [‡] | Lanreotide 120mg every 28days | Placebo | TLV, TKV | 6 months |
| Caroli et al., 2010 | Italy | Multicenter crossover RCT | 12 (75) | 12 (75) | 12 (75) | 44.5 (35–58) [†] | Octreotide 40mg every 28 days | Placebo | TLV, TKV | 6 months |
| Hogan et al., 2010 | USA | Single-center parallel RCT | 42 (14.3) | 28 (17.9) | 14 (7.1) | 49.9 ± 8.38 | Octreotide 40mg every 28 days | Placebo | TLV, TKV, eGFR, QoL | 1 year |
| Caroli et al., 2013 (ALADIN) | Italy | Multicenter parallel RCT | 79 (46.8) | 40 (42.5) | 39 (51.3) | 36.98 ± 8.0 | Octreotide 40mg every 28 days | Placebo | TKV, eGFR | 3 years |
| Pisani et al., 2016 | Italy | Multicenter parallel RCT | 27 (37) | 14 (36) | 13 (38) | 33.37 ± 8.61 | Octreotide 40mg every 28 days | Placebo | TLV | 3 years |
| Meijer et al., 2018 (DIPAK 1) | Netherlands | Multicenter parallel RCT | 305 (46.6) | 153 (46.4) | 152 (46.7) | 48.34 ± 7.29 | Lanreotide 120mg every 28days | Standard care only | eGFR <br> TKV <br> QoL | 2.3 years |
| Perico et al., 2019 (ALADIN 2) | Italy | Multicenter parallel RCT | 100 (57.0) | 51 (60.8) | 49 (53.1) | 49.33 ± 9.07 | Octreotide 40mg every 28 days | Placebo | TKV <br> eGFR | 3 years |
| Van Aerts et al., 2019 | Netherlands | Multicenter parallel RCT | 175 (45.7) | 93 (43.0) | 82 (48.8) | 48.15 ± 6.56 | Lanreotide 120mg every 28 days | Standard care only | htTLV, htTLKV, QoL | 2.3 years |
| Hogan et al., 2020 | USA | Single center randomized clinical trail | 48 (10.4) | 33 (6.1) | 15 (20.0) | 50.55 ± 8.37 | Pasireotide 60mg every 28 days | Placebo | % Change in TLV | 1 year |

[*] Median (IQR).

[†] Median (range).

[‡] 95% CI. RCT = Randomized Controlled Trial. TLV = Total Liver Volume. TKV = Total Kidney Volume. eGFR = estimated Golmerular Filtration Rate. QoL = Quality of Life.

## Outcomes

Our main outcome was to investigate the effect of somatostatin analogs on the TLV, TKV and eGFR in ADPKD or ADPLD patients. TLV and TKV are considered important clinical outcomes in patients with PKD because they are closely related to their quality of life [26, 27]. As the kidney or liver volume may be associated with complications of PKD, such as cyst infection, these outcomes could also predict morbidity and mortality [28]. As the baseline characteristics of the patients are variable (e.g., absolute TLV, TKV and age), we decided to conduct a meta-analysis using change in percentage of TLV and TKV between baseline and follow up (ΔTLV% or ΔTKV%) instead of the absolute value of TLV or TKV, as the change in percentage could be less influenced by the variability in baseline characteristics across studies. Regarding eGFR, we aimed to meta-analyze eGFR as the absolute value since this variable is more standardized across populations.

## Data collection and management

The data from RCTs was extracted using a standardized form. Data extraction was done in a duplicated and independent manner by two reviewers (TS and FJB) after a pilot phase in which reviewers were calibrated. Data on inclusion criteria for each trial, patient demographics, baseline characteristics, sample size, intervention characteristics (type of somatostatin analog and doses), follow-up time, outcome measurements (TLV, TKV, eGFR), and loss to follow-up rate was extracted. If available, the number of events in each trial was extracted and attributed to the arm to which patients were randomized.

## Risk of bias and confidence in the body of evidence

We used the Cochrane risk of bias assessment tool to assess the risk of bias of the primary studies (S2 Fig). This tool takes into consideration seven domains, (1) random sequence generation, (2) allocation concealment, (3) blinding of participants and personnel, (4) blinding of outcome assessment, (5) incomplete outcome data, (6) selective outcome reporting, and (7) other sources of bias. Two reviewers (TS and FJB) independently assessed each study´s quality by examining these domains. Disagreements between the reviewers were resolved by consensus. When reviewers couldn't reach consensus, a third reviewer was consulted (YU or RRG). The overall confidence or overall quality of evidence for each outcome was appraised by discussion between the two extractors using the Grading of Recommendations Assessment, Development and Evaluation (GRADE) approach (S4 Table). This approach takes into account the risk of bias of the individual studies, inconsistency in the results, indirectness, imprecision and other considerations to provide a global assessment of the confidence merited by the body of evidence [29].

## Statistical analysis

**Assumptions and calculations for meta-analysis.** The summary of TLV, TKV and eGFR is shown in Table 2. Six of the 10 articles reported the rate of change in TLV (ΔTLV) and/or TKV (ΔTKV). Perico *et al.* and Temmerman *et al.* reported median and quartile ranges of the rate of change in TLV and TKV (%), we estimated the mean and SD of the rate of change in TLV and TKV under the assumption that the distribution was normal. Meijer *et al.* reported the rate of change in height adjusted TLV and TKV (%), but we considered it was equal to the rate of change in TLV and TKV (%). Caroli *et al.* reported actual TKV of each patient, but they reported only mean and standard deviation (SD) of TKV in their article in 2010. Caroli *et al.* reported mean and standard error (SE) of TKV in their article in 2013. We estimated the rate of change in TKV from these values. We presented the calculation process in S5–S7 Tables.

**Summary measures and data synthesis.** We estimated the mean differences (MD) and SD in rate of ΔTLV, ΔTKV (%), and eGFR (mL/min./1.73m$^2$), and pooled all the studies' effect size using a random-effects model as described by DerSimonian and Kacker [30]. We chose random-effects model as our main method of analysis because of its conservative summary of estimates and incorporation of between- and within-study variability. To assess heterogeneity of treatment effect among trials, we used the I$^2$ statistic; this represents the proportion of heterogeneity of treatment effect across trials that are not attributable to chance or random error. A value of 50% reflects significant heterogeneity and could be due to real differences in study populations, protocols, interventions, or outcomes [31]. Finally, we performed a sensitivity analysis excluding studies with a cross-over design from the meta-analysis to see if the estimate changed because these studies could introduce bias by the presence of a carry-over effect of the intervention. The p value threshold for statistical significance was set at .05 for effect sizes. Analyses were conducted using features on RevMan version 5.3 (The Nordic Cochrane Center,

**Table 2. Total kidney, liver volumes, and estimated glomerular filtration rate reported in (or estimated from) the included studies.**

| Study, year | Follow-up | Somatostatin | | | Control | | |
|---|---|---|---|---|---|---|---|
| | | Pre-treatment | Post-treatment | Estimated or Reported Change % (mean ± SD) | Pre-treatment | Post-treatment | Estimated or Reported Change % (mean ± SD) |
| *Total Liver Volumes (mL)* | | | | | | | |
| LOCKCYST, 2009 | 6 months | 4606 (547–8665)[‡] | 4471 (542–8401)[‡] | -2.9 ± 15.5 | 4689 (613–8765)[‡] | 4896 (739–9053)[‡] | 1.6 ± 12.8 |
| Caroli, 2010 | 6 months | 1595 ± 478 | 1524 ± 453 | -4.03 ± 3.33 | 1580 ± 487 | 1594 ± 480 | 1.23 ± 6.46 |
| Hogan, 2010 | 1 year | 5907.7 ± 2915.0 | 5557.1 ± 2659.4 | -5.0 ± 6.77 | 5373.9 ± 3565.4 | 5360.6 ± 3330.9 | 0.9 ± 8.33 |
| Pisani, 2016 | 3 years | 1609.7 ± 501.2 | 1479.5 ± 470.9 | -7.8 ± 7.4 | 1693.0 ± 470.7 | 1837.2 ± 748.5 | 6.1 ± 14.1 |
| Van Aerts, 2019 | 6 months | 2781 (2272–4230)* | | -1.99 ± 3.30 [¶] | 2389 (2168–3029)* | | 3.92 ± 3.50 [¶] |
| Hogan, 2020 | 1 year | 2582 ± 1381 | 2479 ± 1317 | -3.4 ± 7.3 | 2387 ± 759 | 2533 ± 770 | 6.3 ± 7.0 |
| *Total Kidney Volumes (mL)* | | | | | | | |
| Ruggenenti, 2005 | 6 months | 2551 ± 1053 | 2622 ± 1111 | 2.2 ± 3.7 | 2461 ± 959 | 2623 ± 1021 | 5.9 ± 5.4 |
| LOCKCYST, 2009 | 6 months | 1000 (-39-2039)[‡] | 983 (-62~2028)[‡] | -1.5 ± 31.1 | 1115 (-519~2748)[‡] | 1165 (-541~2871)[‡] | 3.4 ± 28.0 |
| Hogan, 2010 | 1 year | 1142.9 ± 826.9 | 1128.5 ± 796 | 0.25 ± 7.53 | 803 ± 269.1 | 873.5 ± 306.2 | 8.61 ± 10.07 |
| ALADIN, 2013 | 1 year | 1556.9 ± 1035.1[†] | 1603.1 ± 176.1[†] | 2.97 ± 7.41 | 2161.2 ± 1274.9[†] | 2304.9 ± 224.6[†] | 6.65 ± 5.31 |
| | 3 years | 1556.9 ± 1035.1[†] | 1672.7 ± 202.0[†] | 14.14 ± 19.95 | 2161.2 ± 1274.9[†] | 2621.0 ± 271.0[†] | 21.02 ± 32.41 |
| DIPAK 1, 2018 | 1 year | 2046 (1383–2964)* | N/A | 4.15 ± 5.20 | 1874 (1245–2868)* | N/A | 5.56 ± 5.06 |
| ALADIN 2, 2019 | 1 year | 2338.9 (1967.6–3807.4)* | 2513.3 (2023.6–3923.5)* | 5.2 ± 6.37 | 2591.0 (1959.3–3855.7)* | 2935.1 (2197.1–4094.4)* | 8.8 ± 6.15 |
| | 3 years | 2338.9 (1967.6–3807.4)* | 3043.9 (2337.3–5470.6)* | 29.9 ± 21.33 | 2591.0 (1959.3–3855.7)* | 3613.8 (2584.1–4866.8)* | 37.1 ± 23.26 |
| Hogan, 2020 | 1 year | 534 ± 343 | 523 ± 325 | -1.4 ± 3.5 | 397 ± 159 | 417 ± 177 | 3.9 ± 4.5 |
| *Estimated Glomerular Filtration Rate (mL/min./1.73m$^2$)* | | | | | | | |
| Ruggenenti, 2005 | 6 months | 59.5 ± 25.2 | 54.0 ± 23.6 | -5.5 ±10.75 | 57.9 ± 22.49 | 57.7 ± 25.7 | -0.2 ± 7.33 |
| Hogan, 2010 | 1 year | 68.1 ± 26.53 | 64.6 ± 25.66 | -5.1 ± 15.46 | 70.8 ± 28.08 | 65.7 ± 26.40 | -7.2 ± 13.21 |
| ALADIN, 2013 | 1 year | 88.68 ± 3.93[†] | 77.86 ± 4.23[†] | -10.82 ± 7.03 | 77.77 ± 5.30[†] | 72.16 ± 5.45[†] | -5.61 ± 4.09 |
| | 3 years | 88.68 ± 3.93[†] | 76.33 ± 4.66[†] | -3.85 ± 3.17 | 77.77 ± 5.30[†] | 64.64 ± 6.51[†] | -4.95 ± 4.13 |
| DIPAK 1, 2018 | 1 year | 51.0 ± 11.5 | | -3.53 ± 2.93 | 51.4 ± 11.2 | | -3.46 ± 1.39 |
| ALADIN 2, 2019 | 1 year | 27.9 (23.5–32.1)* | 22.5 (17.3~27.7)* | -6.2 ± 2.5 | 25.8 (19.5~33.2)* | 20.2 (14.7~28.1)* | -6.5 ± 0.8 |
| | 3 years | 27.9 (23.5–32.1)* | 14.9 (11.3~20.2)* | -4.26 ± 2.00 | 25.8 (19.5~33.2)* | 15.0 (7.5~24.4)* | -4.19 ± 2.52 |
| Hogan, 2020 | 1 year | 74 ± 24 | 73 ± 22 | -0.3 ± 14 | 74 ± 18 | 73 ± 22 | -2.2 ± 17.7 |

Mean±SD.

*Median (IQR).

[†] mean ± SE.

[‡] 95%CI.

[¶] hight adjusted TLV

Copenhagen, Denmark). Finally, we analyzed the inter-observer agreement in the full text screening phase by calculating Cohen's kappa coefficient.

# Results

## Characteristics of included studies

A total of 10 RCTs that included a total of 854 patients were included in the qualitative and quantitative analysis [32–41]. We excluded the pooled study regarding LOCKCYST I TRIAL

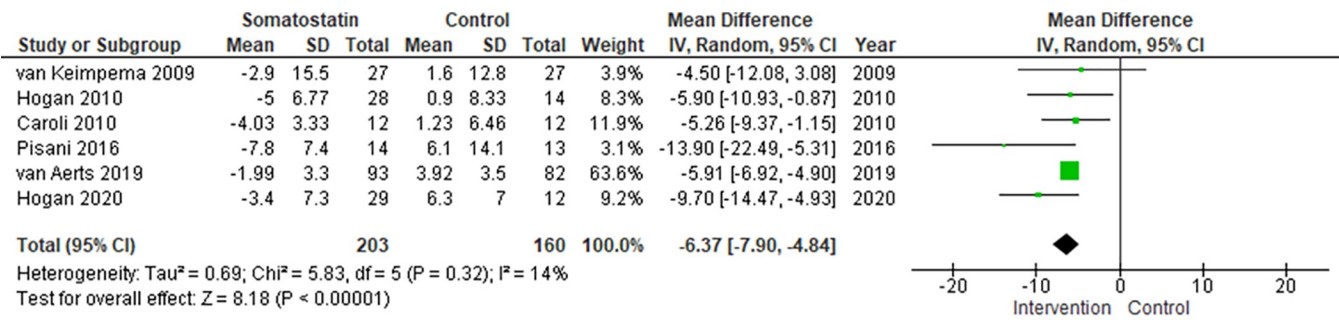

**Fig 2. Meta-analysis of TLV.**

by Temmerman *et al.* in 2013 that overlapped patients with the study by Van Keimpema *et al.* in 2009. The full characteristics of the included studies are summarized in Table 1. Somatostatin analog was administered by intramuscular or subcutaneous injections in all studies; therefore, placebo was not administered in some studies. In these studies, blinding of participants and personnel was not done. Cohen's kappa coefficient resulted in substantial interobserver agreement ($\kappa$ = .71).

## Total Liver Volume (TLV)

On the meta-analysis of 6 studies assessing the effect on TLV (363 patients), Somatostatin analog was associated with lower %TLV growth rate compared to control that was not statistically significant: MD -6.37% (95% CI -7.90 to -4.84, *p<0.00001*; $I^2$ = 14%) (Fig 2).

## Total Kidney Volume (TKV)

On the meta-analysis of 7 studies assessing the effect on TKV (652 patients), Somatostatin analog was significantly associated with lower %TKV growth rate compared to control: MD, -3.66% (95% CI -5.35 to -1.97, *p<0.0001*; $I^2$ = 56%) (Fig 3).

## Estimated Glomerular Filtration Rate (eGFR)

On the meta-analysis of 6 studies assessing the effect on eGFR (576 patients), somatostatin analog showed a slight decrease in eGFR compared to control that was not statistically significant: MD -0.96 mL/min./1.73m$^2$ (95% CI -2.38 to 0.46, *p = 0.19*; $I^2$ = 74%) (Fig 4).

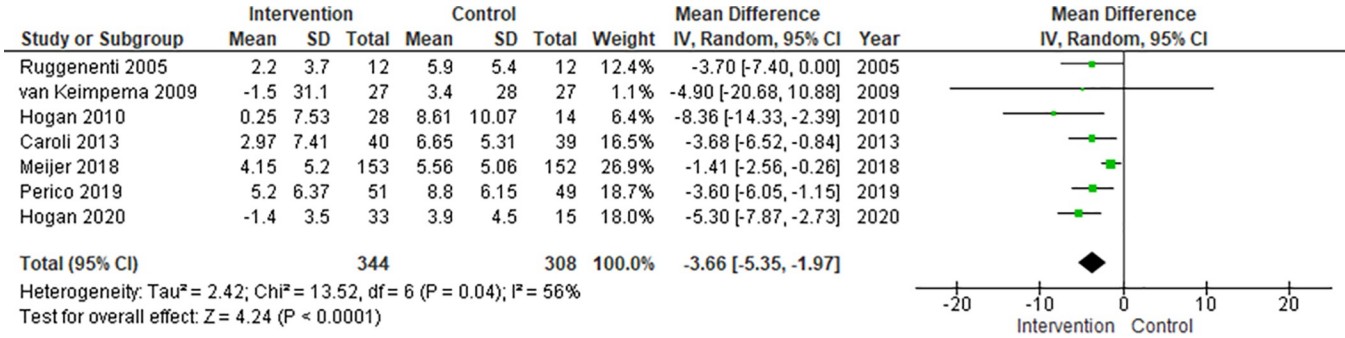

**Fig 3. Meta-analysis of TKV.**

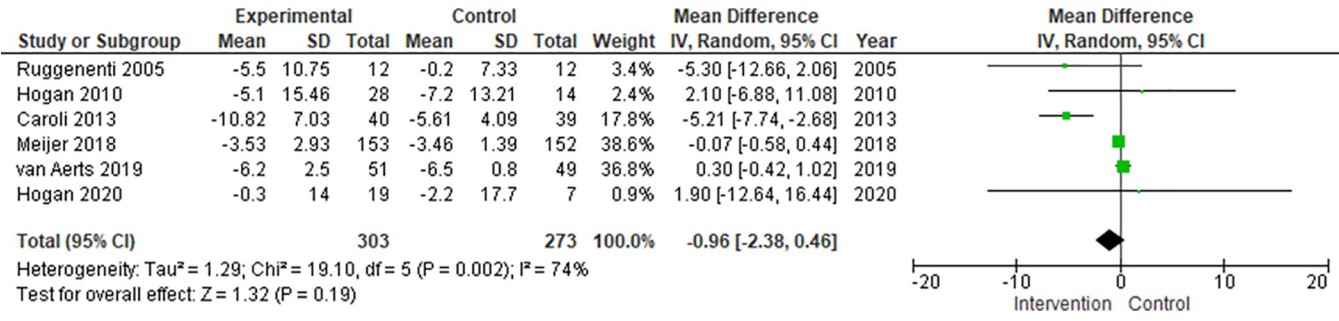

**Fig 4. eGFR.**

## Adverse events

We present the percentage change (frequency in somatostatin group–frequency in control group) of each adverse event. Symptoms such as cholelithiasis/cholecystitis, gastrointestinal symptoms, and hepatic or renal cyst infection were more frequent in somatostatin group than control group. We summarized serious adverse events associated with somatostatin analog in Table 3. All adverse events are presented in S8 Table.

## Sensitivity analyses

In the outcome of TLV, we excluded the study by Caroli et al., 2010. Results suggested that somatostatin analog was still significantly associated with a lower %TKV growth rate compared with control: MD -6.89% (95% CI -9.11 to -4.68, *p<0.00001*; I² = 29%) (S3-1 Fig).

We conducted a sensitivity analysis by study design excluding the studies that had a crossover design because these studies may introduce some bias due to presence of carry-over effect.

**Table 3. Severe adverse events; % in somatostatin group vs. % in control group.**

| Study, year | Biliary complications | | Gastrointestinal complications | | | | Infection | | | Others* |
|---|---|---|---|---|---|---|---|---|---|---|
| | Cholelithiasis | Cholecystitis | Abdominal pain, Epigastric pain, Gastroenteritis | Constipation | Diarrhea | Gastroenteritis, epigastric pain | Hepatic /Renal cyst infection | Hepatic / Renal cyst hemorrhage | Urinary tract infection, pyelonephritis | |
| Ruggenenti, 2005 | 8.3 vs. 0 | N/A | N/A | N/A | 25.0 vs. 0 | N/A | N/A | N/A | N/A | N/A |
| LOCKCYST, 2009 | N/A | N/A | 59.0 vs. 0.0 | 4 vs. 0 | 70.0 vs. 21.0 | N/A | N/A | N/A | N/A | 7.0 vs. 0 |
| Hogan, 2010 | N/A | N/A | 50.0 vs. 21.0 | N/A | 61.0 vs. 28.0 | N/A | N/A | N/A | N/A | N/A |
| ALADIN, 2013 | 5.0 vs. 0 | 5.0 vs. 0 | N/A | N/A | N/A | N/A | | 2.5 vs. 0 | 2.4 vs. 0 | 2.5 vs. 0 |
| Pisani, 2016 | 7.1 vs. 0 | 7.1 vs. 0 | N/A | N/A | N/A | N/A | 7.1 vs. 0 | 14.2 vs. 0 | N/A | 7.1 vs. 0 |
| DIPAK 1, 2018 | 0.7 vs. 0 | N/A | N/A | N/A | N/A | 1.3 vs. 0 | N/A | N/A | N/A | 2.0 vs. 0 |
| Van Aerts, 2019 | 1.1 vs. 0 | N/A | N/A | N/A | N/A | 1.1 vs. 0 | 8.7 vs. 2.2 | N/A | 1.1 vs. 0 | 1.1 vs. 0 |
| ALADIN 2, 2019 | N/A | N/A | N/A | N/A | N/A | N/A | 2.0 vs. 0 | N/A | | 29.4 vs. 16.3 |
| Hogan, 2020 | N/A | N/A | 48.5 vs 40 | 3 vs 6.7 | 51.5 vs 53.3 | N/A | N/A | 3 vs 0 | 9.1 vs 0 | |

* Other all adverse events except for biliary, gastrointestinal, and infectious complications.

In the outcome of TKV, we excluded the studies of Ruggenenti *et al.*, 2005. This demonstrated that somatostatin analog was still significantly associated with a lower %TKV growth rate compared with control: MD -3.73% (95% CI -5.66 to -1.79, *p = 0.0002*; $I^2$ = 62%) (S3-2 Fig).

Finally, in the outcome of eGFR, we excluded the study by Ruggenenti *et al.*, 2009. This showed that somatostatin analog was still not associated with a lower eGFR decline: MD -0.79 mL/min./1.73m$^2$ (95% CI -2.20 to 0.63, *p = 0.28*; $I^2$ = 77%) (S3-3 Fig).

## Risk of bias and confidence in the overall body of evidence

All of the studies included a low rate of lost to follow-up, representing low risk of attrition bias in some RCTs (S9 Table, S2 Fig). Random sequence generation and allocation concealment were graded as low risk of bias for the majority of the studies. Blinding of participants and personnel represented a concern in 5 studies. However, outcome assessment was blinded in almost all of the studies. Finally, studies using a crossover design might have introduced bias through the "carryover effect" but this was analyzed through a sensitivity analysis, and we concluded these studies were unlikely to introduce bias through this phenomenon because the estimates did not change significantly.

The overall quality of the evidence was considered high for the efficacy in TLV (S4 Table); and moderate for TKV due to inconsistency of results. Regarding the eGFR outcome, quality of the evidence was graded as low due to inconsistency of results and indirectness. The result of risk of bias assessment is presented in S2 Fig.

## Discussion

### Main findings

We found a significant reduction in change in TLV and TKV. This difference was relatively large (6.37% for ΔTLV and 3.66% for ΔTKV). We consider that this should be considered as a significant clinical benefit to the patients. We observed no difference in the eGFR decline; our analysis suggested a high heterogeneity for this outcome which could be partially explained by a different direction of the effect estimates in the included studies.

### Comparison with previous studies

Compared with a previous meta-analysis, we estimated the effect of somatostatin analogues on TLV or TKV using the change in percentage instead of absolute values [22]. We considered this methodology more appropriate because baseline TLV or TKV is highly heterogeneous among studies and therefore the analysis could be over- or under-estimated, and this could lead to inaccurate efficacy estimates. Other previous meta-analyses have concluded controversial results regarding the efficacy of this intervention. Our results resonate with those of two previous meta-analysis that found no difference in the eGFR decline rate after the intervention. Conversely, our results differ from those in a previous analysis that suggested that this intervention shows no benefit in TKV. We consider that our meta-analysis was more sensitive to detect the effectiveness of somatostatin analogues on TLV and TKV in comparison to the previous analysis that used the absolute values of TLV and TKV because we used the change in percentage of TLV and TKV for the analysis. This suggests that somatostatin analogs are more likely to be effective for early-stage patients or patients with a smaller TLV or TKV.

### Implications for clinical practice and research

TLV and TKV are considered important clinical outcomes in patients with PKD because they are closely related to their quality of life, morbidity and mortality [26–28]. Our results suggest

that somatostatin analogs are effective for slowing growth of PKD or PLD, however the treatment effect of somatostatin analogs could vary among individual patients. The average age of the enrolled patients in the study by Pisani 2020 was the youngest in all studies and the effect of somatostatin analogs on TLV was the strongest among all studies. Meijer et al. also reported that somatostatin analogs were more effective for patients whose age (≤45) than those whose age (>45) though it was not statistically significant [38]. Among the enrolled RCTs of this meta-analysis, van Aerts *et al.*, presented a subgroup analysis that suggested that patients ≤ 45 years old seemed to have more benefits from somatostatin analogs compared to those > 45, however this effect was not statistically significant [40]. These suggests that somatostatin analogs are more effective on younger patients, which was consistent with the report by Gevers *et al.*, They reported that young female patients (48 years old and younger) seemed to have the most substantial effect of somatostatin analogs in a pooled analysis [42]. Some studies reported multiple pregnancies and exogenous estrogens as risk factors for growth of hepatic cysts [43, 44]. Gevers et al. mentioned that premenopausal status may be an independent risk factor for polycystic liver growth due to hormonal influence [42]. Cholangiocyte proliferation is considered one of the major contributors to hepatic cystogenesis and is significantly increased by estrogens in vitro [45–47]. Thus, liver cysts grow rapidly in young women and somatostatin analogs may be the more effective for such patients with extensive cyst proliferation. It has also been suggested that estrogens may enhance the ability of somatostatin analogs to inhibit cyclic adenosine monophosphate production in cholangiocytes, and can increase susceptibility to somatostatin analogs therapy in fertile women [42]. Taken together, we hypothesize that young women may receive the most benefit from somatostatin analogs; however, the primary studies lack subgroup analyses. Moreover, our results suggest that TKV seems to be less effected and eGFR does not seem to be affected by somatostatin analog therapy. Nonetheless, longer follow-up periods could be useful to further clarify any effectiveness on TKV and eGFR.

The frequency of reported adverse events with somatostatin analog therapy was high in all RCTs. However, the rate of adverse events seems to be almost same as the rates reported in other clinical trials using somatostatin analogs for other diseases, such as acromegaly and neuroendocrine tumors [48].

End stage renal disease (ESRD) and death may be the most important outcomes. The study by Perico et al., reported that 3 patients in the intervention group (5.9%) progressed to end stage renal disease compared to 8 (16.9%) in the placebo group. In the study of Meijer et al., 1 patient in the intervention group died but they report that this patient was diagnosed with lung cancer during the study. However, only 4 studies (Meijer et al., Perico et al., and Hogan et al.) considered ESRD as an outcome. Additionally, death was reported by 2 studies (Meijer et al. and Van Aerts et al.), none of the studies considered it as an outcome S10 Table. As such, there is scarcity of information regarding these patient-important outcomes and therefore, we were unable to assess the effectiveness of somatostatin analogs on ESRD or death. PKD or PLD are slowly progressive diseases and longer study periods may be necessary to evaluate these outcomes.

Basic optimized treatments for ADPKD include rigorous blood pressure control and various dietary changes [49]. Disease modifying treatment for ADPKD is currently very limited, but tolvaptan (a vasopressin V2 receptor antagonist) has been approved in several countries. According to the TEMPO 3:4 study, the rate of any adverse events was 97.9% among patients who received tolvaptan and a total of 15.4% of the patients who received tolvaptan permanently discontinued the trial drug due to adverse events associated with the drug [50]. The common adverse events of tolvaptan are thirsty and polyuria. On the other hand, the common adverse events of somatostatin analogs are digestive problems and the rate of discontinuation

in patients who received somatostatin analogs as a trial drug was up to 15%. Taken together, the tolerability of somatostatin analogs may not be worse than that of tolvaptan.

Cost for somatostatin analogs is as high as that for tolvaptan ($8,011 for tolvaptan, $7,960 for 40mg octreotide LAR, and $10,144 for 120mg lanreotide per month in the U.S.). However, as there are no studies head-to-head trials comparing somatostatin analog and tolvaptan, we were not able to compare tolvaptan versus somatostatin analogs directly. Tolvaptan is generally ineffective for slowing progression of PLD because vasopressin V2-receptor is unique for kidney, therefore somatostatin analog may be the only current available drug to slow progression of PLD. In that sense, somatostatin analog use could be more justified in patients with PLD. Moreover, somatostatin analogs and tolvaptan could provide synergistic effect if used together. If true, the required dose of somatostatin analogs could be reduced if used along with tolvaptan. Nonetheless, further studies are necessary to clarify the relationship between somatostatin analogs and tolvaptan.

Ultimately, we consider that somatostatin analogs could be effective even though the disadvantages -relatively high adverse event frequency and high therapy cost- it may carry. However, more studies are needed to further define which particular patients could experience the greatest benefit by this therapy.

## Strengths and limitations

The systematic approach of this review and the thorough search strategy strengthens our study. Moreover, the moderate confidence of our estimates and the fact that the average age of patients in 7 of 10 studies was similar, also provide strength to our results.

There are also some limitations in this study. First, the follow-up periods were variable (ranged from 6 months to 3 years). This difference in follow up time might have affected the results of our meta-analysis. The baseline characteristics of enrolled patients were also variable among each study. For example, baseline TLV and TKV were variable even though they may be important predictive factors. The average TKV ranged from 1000 to 2600 mL. The range of mean TLV of included studies was more variable. As such, the mean TLV was about 1590 mL in the study by Caroli *et al.* in 2010, whereas that in the study by Hogan *et al.* in 2010 was about 5900 mL. However, it seems that there was not any association between baseline TLV/TKV and the effectiveness of the intervention. In addition, other important baseline characteristics on enrolled patients such as genetics characteristics, blood pressure and concomitant medications are unknown in many studies. Moreover, some studies included patients with PLD instead of PKD. Also, two of the 10 RCTs had crossover design. Additionally, since all of the studies included in this meta-analysis were conducted in Europe or the U.S. and the included patients were mostly Caucasians, we cannot truly generalize these results to the other races or other regions of the world. Besides, the number of included studies is not large, and this made it difficult to evaluate the risk of publication bias. At last, as the number of studies included in this meta-analysis is limited, accumulation of studies would be necessary to make stronger conclusions.

## Conclusion

The body of evidence shows that somatostatin analog therapy slows increase rate of TLV and TKV in patients with PLD or PKD compared to control group within a 3-year follow-up period. However, somatostatin analogs are associated with severe adverse events and high costs. More evidence is needed to further define to which patients could this therapy be justified, and which patients would receive the greatest benefit from it.

## Supporting information

**S1 Fig. PRISMA 2009 flow diagram.**
(DOC)

**S2 Fig. Risk of bias assessment.**
(DOCX)

**S3 Fig. Sensitivity analysis.**
(DOCX)

**S1 Table. PRISMA 2009 checklist.**
(DOC)

**S2 Table. Search strategy.**
(DOCX)

**S3 Table. List of excluded studies.**
(DOCX)

**S4 Table. Summary of findings and confidence in the body of evidence.**
(DOCX)

**S5 Table. calculation process to estimate ΔTLV (mean±SD) %.**
(DOCX)

**S6 Table. Calculation process to estimate ΔTKV (mean±SD) %.**
(DOCX)

**S7 Table. Calculation process to estimate eGFR (mean±SD) mL/min./1.73m2.**
(DOCX)

**S8 Table. List of all adverse events.**
(DOC)

**S9 Table. Loss to follow up.**
(DOCX)

**S10 Table. Hard outcome.**
(DOCX)

## Acknowledgments

TS and FJB were scholars of the class of Systematic Reviews and Meta-Analysis in Mayo Clinic's Graduate School, which was directed by: Victor Montori, MD, MSc; Colin West, MD, PhD; and M. Hassan Murad, MD in 2018, and this study is followed by the policy of this class. Analysis and interpretation of data in this study was supported by Satista (Kyoto, Japan). We thank Ms. Lisa E. Vaughan for providing us the data about the study by Dr. Hogan in 2020.

## Author Contributions

**Conceptualization:** Tatsuya Suwabe, Francisco J. Barrera.

**Data curation:** Tatsuya Suwabe, Francisco J. Barrera.

**Formal analysis:** Tatsuya Suwabe, Francisco J. Barrera.

**Funding acquisition:** Tatsuya Suwabe.

**Investigation:** Tatsuya Suwabe, Francisco J. Barrera.

**Methodology:** Tatsuya Suwabe, Francisco J. Barrera.

**Project administration:** Tatsuya Suwabe, Francisco J. Barrera.

**Resources:** Tatsuya Suwabe, Francisco J. Barrera.

**Software:** Tatsuya Suwabe, Francisco J. Barrera.

**Supervision:** Rene Rodriguez-Gutierrez, Yoshifumi Ubara, Marie C. Hogan.

**Validation:** Tatsuya Suwabe, Francisco J. Barrera, Rene Rodriguez-Gutierrez, Marie C. Hogan.

**Visualization:** Tatsuya Suwabe, Francisco J. Barrera, Rene Rodriguez-Gutierrez, Yoshifumi Ubara, Marie C. Hogan.

**Writing – original draft:** Tatsuya Suwabe.

**Writing – review & editing:** Francisco J. Barrera, Rene Rodriguez-Gutierrez, Yoshifumi Ubara, Marie C. Hogan.

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
