## [Decision Letter · Decision Letter 0]

26 May 2021

PONE-D-21-10078

Somatostatin Analog Therapy Effectiveness on the Progression of Polycystic Kidney and Liver Disease: A Systematic Review and Meta-analysis of Randomized Clinical Trials

PLOS ONE

Dear Dr. Suwabe,

Thank you for submitting your manuscript to PLOS ONE. After careful consideration, we feel that it has merit but does not fully meet PLOS ONE’s publication criteria as it currently stands. Therefore, we invite you to submit a revised version of the manuscript that addresses the points raised during the review process.

**The manuscript focuses on a topic of potential interest. Some shortcomings, however, should be addressed. To mention few of them, i) unclear in the Methods what the authors exactly want to investigate and compare between the different trials in their meta-analysis; ii) unclear how do they define effectiveness, and what are the most important outcomes, information to be added in the Methods; iii) need to add more speculations in the Discussion section; iv) need to reorganize the abstract; v) concern about the fact that they mention in the Methods that they included RCTs that compared a somatostatin analogue versus placebo, although many of the considered studies did not include a placebo group; vi) need to mention the ‘hard outcomes’ in the Discussion section instead of in the Results section; vii) unclear what they mean as ‘change’ for the adverse events; viii)  unclear how and the reasons to perform several sensitivity analyses; ix) unclear (in the discussion) why female patients seemed to have the most substantial effect of somatostatin analogues in a polled analysis; x) unclear why the somatostatin analogues are better tolerated in PKD and PLD than in acromegaly and neuroendocrine tumours; xi) need to re-write the sentence 275; xii) need to revise tables and figures as suggested by Reviewers 2; xiii) need to know which pharmacological therapies were used and which medications were not allowed; xiv) need to perform some explorative subgroup analyses considering early vs late CKD stages; xv) need to consider that not all the main efficacy variables were primary; xvi) need to mention in the limitations of the study, the fact that estimated GFR can be unreliable.**

We look forward to receiving your revised manuscript.

Kind regards,

Giuseppe Remuzzi

Academic Editor

PLOS ONE

Journal Requirements:

"This work was supported by JSPS KAKENHI Grant Number JP19K17758. This study was

329 supported in part by a Grant-in-Aid for Progressive Renal Disease Research from the

330 Ministry of Health, Labour and Welfare of Japan and by Okinaka Memorial Institute for

331 Medical Research, Toranomon Hospital.2

"The authors received no specific funding for this work."

3. Please include a copy of Table 5 which you refer to in your text on page 194.

Reviewers' comments:

Reviewer's Responses to Questions

**Comments to the Author**

1. Is the manuscript technically sound, and do the data support the conclusions?

Reviewer #1: Yes

Reviewer #2: Yes

2. Has the statistical analysis been performed appropriately and rigorously? 

Reviewer #1: Yes

Reviewer #2: Yes

3. Have the authors made all data underlying the findings in their manuscript fully available?

Reviewer #1: Yes

Reviewer #2: Yes

4. Is the manuscript presented in an intelligible fashion and written in standard English?

Reviewer #1: Yes

Reviewer #2: Yes

5. Review Comments to the Author

Reviewer #1: Summary of paper

The Authors carried out a systematic review and meta-analysis based on seven randomized trials involving 652 participants overall, in order to assess the efficacy and safety of somatostatin analogues in ADPKD. They found that use of somatostatin analogues was associated with a significantly decrease in TKV and TLV than placebo, but no difference in estimated GFR decline was observed. This manuscript covers an important topic. It is generally well designed and reported: some minor comments are listed below.

Abstract

- The Authors should specify the median follow up length of the included studies.

Eligibility criteria for studies

- The ‘control’ group was clearly identified, i.e. ‘placebo’. However the use of concomitant medication at randomization may differ from study to study. It can be useful to know which pharmacological therapies could be used (e.g. ACE inhibitors, angiotensin receptor blockers and other anti-hypertensive agents, statins, …) and which medications were not allowed.

Results

- Some exploratory subgroup analyses can be undertaken considering early vs late CKD stages (e.g. eGFR above and below 40 ml/min/1.73 m2, or according to 24h proteinuria categories). Although these findings are to be considered ancillary, they could be helpful for data interpretation and for assessing generalizability of study findings.

Discussion

- Maybe not all the main efficacy variables were ‘primary’: this aspect should be taken into account when interpreting results from secondary efficacy variables, which are more prone to publication bias.

- GFR decline was evaluated based on estimated GFR only. Failure to demonstrate protective effects of somatostatin analogues on GFR in ADPKD can be (at least partially) explained by unreliable GFR estimates. This aspect should be mentioned among the limitations of the study.

Reviewer #2: In this well written systematic review and meta-analysis, Suwabe et al. summarizes the most important RCT’s that investigate the effectiveness of somatostatin analogues to slow progression of PKD and PLD. However, I think some of the information can be written or presented more clearly for the reader. Furthermore I miss some novelty of this review. Therefore I have a few suggestions for improvement.

Major comments

- The description on how the selection for the RCTs have been conducted is very clear. However, I miss in the methods what the authors exactly want to investigate and compare between the different trials in their meta-analysis. How do they define effectiveness? What are the most important outcomes? That only becomes clear in the result section. I think it would be more clear for the reader when the authors add more about this in their method section.

- I miss some speculation and discussion about some of the results in de discussion section. This research group is very specialized in PLD and somatostatin analogues and I am very curious about their thought about some of the results. This manuscript would be more novel when the authors add more speculation in their discussion section.

Minor comments

Abstract

- The authors mention in the result section first TKV and then TLV numbers. At last eGFR is mentioned. I would first describe TLV, then TKV and subsequently eGFR data. This is a more logical order because ofcourse TKV and eGFR are related.

- In the conclusion I miss a few words about ‘suggestions for the future’

Introduction

- The authors mention a good point about results with absolute volumes versus percentage change. This is very important and very often wrongly interpreted by some authors.

Methods

- The authors mention that they included RCTs that compared a somatostatin analogue versus placebo, although many of the included studies did not include a placebo group. Please change this wording.

- As mentioned in the major comments, I miss the description of which outcomes the authors want to investigate in their meta-analysis.

Results

- I understand that he authors want to say something about endpoints and somatostatin analogues. However, the data mentioned under ‘Hard outcomes’ is in my opinion not relevant. I think it would be better not mention this data here, but describe this in the discussion section for example. Also the authors should add some wording about what they think the reason is that there are no hard outcomes.

- Adverse events are presented as ‘change’; but change from what? Before and after starting somatostatin analogues? Or is this just a comparison between patients using somatostatin analogues and not using somatostatin analogues? In that case, change is in my opinion not the correct way to describe this data. Please only use percentage vs. percentage between two groups.

- The authors describe that they have performed several sensitivity analysis; do they mean that they only excluded studies with a cross-over design? First of all, isn’t that just one sensitivity analysis instead of several? Furthermore for me it is not quite clear why the authors perform this analysis; I think some additional wording should be added to explain this.

Discussion

- This is very minor, but I noticed two small typo’s:

Sentence 227 ‘ We consider that it this should be’

Sentence 250 ‘for patients whose age (<45) that those whose age’

- In sentence 253 the authors mention that young female patients seemed to have the most substantial effect of somatostatin analogues in a polled analysis. I miss here the explanation what the reason for this is. I am sure the authors are aware of de decrease in liver growth in women after menopause.

- Also very minor: Sentence 255 the authors ‘Rene et al’ are mentioned; this is his first name, his last name is ‘van Aerts’.

- In sentence 259 and on, the authors mention that they could not elucidate which subgroups could get the highest benefit from this therapy. Can the authors maybe speculate about this? Maybe young women are the ones who have the most benefit in term of absolute growth in liver volume? And maybe the fact that they do not need a liver transplant? The letter is also nowhere mentioned in this manuscript, although an important point.

- About adverse events of the somatostatin analogues, what do the authors think is the reason that they are better tolerated in PKD and PLD than in acromegaly and neuroendocrine tumors?

- In sentence 275 tolvaptan is suddenly mentioned. I ofcourse understand why tolvaptan is metioned, but it looks that it is mentioned here ‘out of the blue’. I think the authors should first begin that there is another therapy available, what it is, and then start to compare tolvaptan with somatostatin analogues. I think also important is maybe to compare adverse events om somatostatin analogues with tolvaptan. In my clinical experience somatostatin analogues are better tolerated than tolvaptan (but I believe existing data about this does not represent my experience).

Conclusion

- Here it is mentioned that somatostatin analogues slow progression of PKD and PLD; I think it would be better to mention here that there is only volume reduction…..

Tables

Table 1

- The title of the 5th column is ‘Somatostatin’ the 6th ‘Control’. Please adjust this to for example ‘Somatostatin therapy’ and ‘Control group’, to make this more clear.

- The 9th column also mentiones ‘Control’ but there are no data for this column. Please delete this column.

Table 2

- There are no numbers for Post-treatment mentioned for ‘DIPAK 1’ and ‘van Aerts’. Is that data not available or is this by accident not filled out? Please clarify in the table when it is not available with for example putting NA in those fields.

Table 3

- As mentioned before, I would advise to only mention the percentages for each group instead of making a variable that represents change (which is in my opinion not suitable here).

Figures

Figure 2

- Also mention in the figures somatostatin therapy instead of only somatostatin to make it clear that you are talking about intervention versus control group.

Figure 3

- For the results of ‘van Aerts’ the forest plot shows a large green square and no 95% CI. This seems a bit odd. Is this an error in the figure?

6. PLOS authors have the option to publish the peer review history of their article (what does this mean?). If published, this will include your full peer review and any attached files.

Reviewer #1: **Yes: **Annalisa Perna

Reviewer #2: No

---

## [Author Response · Author response to Decision Letter 0]

14 Aug 2021

We have extensively revised the manuscript according to the comments and suggestions of the Editors and Referees. We greatly appreciate these comments and suggestions which we believe have resulted in an improved manuscript. Point-by-point responses to the editors' and referees' comments are listed below. 

Major comments

The description on how the selection for the RCTs have been conducted is very clear. However, I miss in the methods what the authors exactly want to investigate and compare between the different trials in their meta-analysis. How do they define effectiveness? What are the most important outcomes? That only becomes clear in the result section. I think it would be more clear for the reader when the authors add more about this in their method section. 

→　Our main outcome was to investigate the effect of somatostatin analogs on the TLV, TKV and eGFR in ADPKD or ADPLD patients. TLV and TKV are considered important clinical outcomes in patients with PKD because they are closely related to their quality of life [A,B]. As the kidney or liver volume may be associated with complications of PKD, such as cyst infection, these outcomes could also predict morbidity and mortality [C]. As the baseline characteristics of the patients are variable (e.g., absolute TLV, TKV and age), we decided to conduct a meta-analysis using change in percentage of TLV and TKV between baseline and follow up (ΔTLV% or ΔTKV%) instead of the absolute value of TLV or TKV, as the change in percentage could be less influenced by the variability in baseline characteristics across studies. Regarding eGFR, we aimed to meta-analyze eGFR as the absolute value since this variable is more standardized across populations.

We added a new paragraph “Outcomes” in the “Methods” section. We also added some comments in the “Comparison with previous studies“ section. 

Reference

A. Miskulin DC, Abebe KZ, Chapman AB, Perrone RD, Steinman TI, Torres VE, et al. Health-related quality of life in patients with autosomal dominant polycystic kidney disease and CKD stages 1-4: a cross-sectional study. Am J Kidney Dis. 2014;63(2):214-26. 

B. Suwabe T, Ubara Y, Mise K, Kawada M, Hamanoue S, Sumida K, et al. Quality of life of patients with ADPKD-Toranomon PKD QOL study: cross-sectional study. BMC Nephrol. 2013;14:179.

C. Suwabe T, Ubara Y, Hayami N, Yamanouchi M, Hiramatsu R, Sumida K, et al. Factors Influencing Cyst Infection in Autosomal Dominant Polycystic Kidney Disease. Nephron.

I miss some speculation and discussion about some of the results in de discussion section. This research group is very specialized in PLD and somatostatin analogues and I am very curious about their thought about some of the results. This manuscript would be more novel when the authors add more speculation in their discussion section.

→　We consider that our meta-analysis found more difference in the effectiveness on TLV and TKV in comparison to the previous analysis that used the absolute values of TLV and TKV because we used the rate of change in TLV and TKV for the analysis. This suggests that somatostatin analogs are more likely to be effective for early-stage patients or patients with a smaller TLV or TKV. In addition, since some studies reported that multiple pregnancies and exogenous estrogens are risk factors for the growth of hepatic cysts (D, E), Gevers et al. mentioned that a premenopausal status may be an independent risk factor for polycystic liver growth due to hormonal influence. Cholangiocyte proliferation is considered to be one of the major contributors to hepatic cystogenesis and is reported to be significantly increased by estrogens in vitro (F, G, H). Thus, liver cyst growth may be rapid in young women and somatostatin analogs may be more effective for such patients with extensive cyst proliferation. It has also been suggested that estrogens, apart from increasing cholangiocyte proliferation, may enhance the ability of somatostatin analogs to inhibit cyclic adenosine monophosphate production in cholangiocytes, and that they can increase susceptibility to somatostatin analog therapy in fertile women (I).

References

D. A.B. Chapman Cystic disease in women: clinical characteristics and medical management. Adv Ren Replace Ther, 10 (2003), pp. 24-30

E. P.A. Gabow, A.M. Johnson, W.D. Kaehny, et al. Risk factors for the development of hepatic cysts in autosomal dominant polycystic kidney disease. Hepatology, 11 (1990), pp. 1033-1037

F. M. Strazzabosco, S. Somlo. Polycystic liver diseases: congenital disorders of cholangiocyte signaling. Gastroenterology, 140 (2011), pp. 1855-1859

G. D. Alvaro, G. Alpini, P. Onori, et al. Estrogens stimulate proliferation of intrahepatic biliary epithelium in rats. Gastroenterology, 119 (2000), pp. 1681-1691

H. D. Alvaro, P. Onori, G. Alpini, et al. Morphological and functional features of hepatic cyst epithelium in autosomal dominant polycystic kidney disease. Am J Pathol, 172 (2008), pp. 321-332

I. 42. Gevers TJ, Inthout J, Caroli A, Ruggenenti P, Hogan MC, Torres VE, et al. Young women with polycystic liver disease respond best to somatostatin analogues: a pooled analysis of individual patient data. Gastroenterology. 2013;145(2):357-65.e1-2. 

Minor comments

Abstract

- The authors mention in the result section first TKV and then TLV numbers. At last eGFR is mentioned. I would first describe TLV, then TKV and subsequently eGFR data. This is a more logical order because ofcourse TKV and eGFR are related.

→　We changed the order throughout this manuscript.

- In the conclusion I miss a few words about ‘suggestions for the future’ 

→　We added the following sentence: “Somatostatin analog therapy can be a promising treatment for ADPKD or ADPLD, and we need to continue to research its effectiveness for ADPKD or ADPLD.”

Introduction

The authors mention a good point about results with absolute volumes versus percentage change. This is very important and very often wrongly interpreted by some authors.

→　Thank you very much for your comment. I agree with your opinion. We modified a sentence in the “Introduction” section (page 5). 

Methods

The authors mention that they included RCTs that compared a somatostatin analogue versus placebo, although many of the included studies did not include a placebo group. Please change this wording.

→　 We changed the word “placebo” to “control”. (Page 5)

As mentioned in the major comments, I miss the description of which outcomes the authors want to investigate in their meta-analysis. 

→　We added a new paragraph “Outcomes” in the “Methods” section.

Results

I understand that he authors want to say something about endpoints and somatostatin analogues. However, the data mentioned under ‘Hard outcomes’ is in my opinion not relevant. I think it would be better not mention this data here, but describe this in the discussion section for example. Also the authors should add some wording about what they think the reason is that there are no hard outcomes.

→　We moved these sentences about ‘Hard outcomes’ to the “Discussion” section. PKD or PLD are slowly progressive diseases and longer study periods may be necessary to evaluate these outcomes.

Adverse events are presented as ‘change’; but change from what? Before and after starting somatostatin analogues? Or is this just a comparison between patients using somatostatin analogues and not using somatostatin analogues? In that case, change is in my opinion not the correct way to describe this data. Please only use percentage vs. percentage between two groups.

→　We used only percentage vs. percentage for comparisons between the two groups.

The authors describe that they have performed several sensitivity analysis; do they mean that they only excluded studies with a cross-over design? First of all, isn’t that just one sensitivity analysis instead of several? Furthermore for me it is not quite clear why the authors perform this analysis; I think some additional wording should be added to explain this.

→ 　We introduced the reasons for this and changed the wording for a single sensitivity analysis: “We conducted a sensitivity analysis by study design excluding the studies that had a cross-over design because these studies may introduce some bias due to presence of carry-over effect”

We also added the following in the statistical analysis section of the methods: “Finally, we performed a sensitivity analysis excluding studies with a cross-over design from the meta-analysis to see if the estimate changed because these studies could introduce bias by the presence of a carry-over effect of the intervention.” 

Discussion

This is very minor, but I noticed two small typo’s:

Sentence 227 ‘ We consider that it this should be’ 

Sentence 250 ‘for patients whose age (<45) that those whose age’ 

→　Thank you for pointing out our mistakes. We corrected the sentences.

In sentence 253 the authors mention that young female patients seemed to have the most substantial effect of somatostatin analogues in a polled analysis. I miss here the explanation what the reason for this is. I am sure the authors are aware of de decrease in liver growth in women after menopause. 

→　Some studies reported multiple pregnancy and exogenous estrogens as risk factors for the growth of hepatic cysts (A, B). Gevers TJ, et al. mentioned that a premenopausal status may be an independent risk factor for polycystic liver growth due to hormonal influence. Liver cysts grow rapidly in young women and somatostatin analogs may be the more effective for such patients with extensive cyst proliferation.

References

D) A.B. Chapman. Cystic disease in women: clinical characteristics and medical management

Adv Ren Replace Ther, 10 (2003), pp. 24-30.

E) P.A. Gabow, A.M. Johnson, W.D. Kaehny, et al. Risk factors for the development of hepatic cysts in autosomal dominant polycystic kidney disease. Hepatology, 11 (1990), pp. 1033-1037.

Also very minor: Sentence 255 the authors ‘Rene et al’ are mentioned; this is his first name, his last name is ‘van Aerts’.

→　Thank you for telling us the mistake. We corrected the sentence.

In sentence 259 and on, the authors mention that they could not elucidate which subgroups could get the highest benefit from this therapy. Can the authors maybe speculate about this? Maybe young women are the ones who have the most benefit in term of absolute growth in liver volume? And maybe the fact that they do not need a liver transplant? The letter is also nowhere mentioned in this manuscript, although an important point.

→　We corrected the sentences as follows: Taken together, we hypothesize that young women may receive the most benefit from somatostatin analogs; however, the primary studies lack subgroup analyses.

About adverse events of the somatostatin analogues, what do the authors think is the reason that they are better tolerated in PKD and PLD than in acromegaly and neuroendocrine tumors?

→　We corrected the sentences. The rate of adverse events seems to be almost the same as the rates reported in other clinical trials using somatostatin analogs for other diseases, such as acromegaly and neuroendocrine tumor. 

In sentence 275 tolvaptan is suddenly mentioned. I of course understand why tolvaptan is metioned, but it looks that it is mentioned here ‘out of the blue’. I think the authors should first begin that there is another therapy available, what it is, and then start to compare tolvaptan with somatostatin analogues. I think also important is maybe to compare adverse events om somatostatin analogues with tolvaptan. In my clinical experience somatostatin analogues are better tolerated than tolvaptan (but I believe existing data about this does not represent my experience).

→　Thank you for your suggestion. We added some comments regarding the general treatment of ADPKD as follows.

Basic optimized treatments for ADPKD include rigorous blood pressure control and various dietary changes. Disease modifying treatment for ADPKD is currently very limited, but tolvaptan (a vasopressin V2 receptor antagonist) has been approved in several countries. 

According to the TEMPO 3:4 study, the rate of any adverse events was 97.9% among patients who received tolvaptan and a total of 15.4% of the patients who received tolvaptan permanently discontinued the trial drug due to adverse events associated with the drug. The common adverse events of tolvaptan are thirsty and polyuria. On the other hand, the common adverse events of somatostatin analogs are digestive problems and the rate of discontinuation in patients who received tolvaptan as a trial drug was up to 15%. Taken together, the tolerability of somatostatin analogs may not worse than that of tolvaptan.

Conclusion

Here it is mentioned that somatostatin analogues slow progression of PKD and PLD; I think it would be better to mention here that there is only volume reduction…..

→　We modified the sentence according to your suggestion.

Tables

Table 1

The title of the 5th column is ‘Somatostatin’ the 6th ‘Control’. Please adjust this to for example ‘Somatostatin therapy’ and ‘Control group’, to make this more clear.

→　We adjusted this to make it clearer.

The 9th column also mentiones ‘Control’ but there are no data for this column. Please delete this column.

→　We included the types of therapy for the control group in this column.

Table 2

There are no numbers for Post-treatment mentioned for ‘DIPAK 1’ and ‘van Aerts’. Is that data not available or is this by accident not filled out? Please clarify in the table when it is not available with for example putting NA in those fields.

→　They did not report those data. We inserted “N/A” in the fields.

Table 3

As mentioned before, I would advise to only mention the percentages for each group instead of making a variable that represents change (which is in my opinion not suitable here).

→　We have modified the table according to your suggestion.

Figures

Figure 2

Also mention in the figures somatostatin therapy instead of only somatostatin to make it clear that you are talking about intervention versus control group. 

→　We changed the subtitles of the figures.

Figure 3

- For the results of ‘van Aerts’ the forest plot shows a large green square and no 95% CI. This seems a bit odd. Is this an error in the figure?

→　It is not a mistake. The number of patients was large and the 95% CI was small.

---

## [Decision Letter · Decision Letter 1]

7 Sep 2021

Somatostatin Analog Therapy Effectiveness on the Progression of Polycystic Kidney and Liver Disease: A Systematic Review and Meta-analysis of Randomized Clinical Trials

PONE-D-21-10078R1

Dear Dr. Suwabe,

We’re pleased to inform you that your manuscript has been judged scientifically suitable for publication and will be formally accepted for publication once it meets all outstanding technical requirements.

**The revised manuscript is definitely improved. The authors have properly addressed the comments raised by the reviewers.**

Kind regards,

Giuseppe Remuzzi

Academic Editor

PLOS ONE

Additional Editor Comments (optional):

Reviewers' comments:

Reviewer's Responses to Questions

**Comments to the Author**

1. If the authors have adequately addressed your comments raised in a previous round of review and you feel that this manuscript is now acceptable for publication, you may indicate that here to bypass the “Comments to the Author” section, enter your conflict of interest statement in the “Confidential to Editor” section, and submit your "Accept" recommendation.

Reviewer #1: All comments have been addressed

Reviewer #2: All comments have been addressed

2. Is the manuscript technically sound, and do the data support the conclusions?

Reviewer #1: Yes

Reviewer #2: Yes

3. Has the statistical analysis been performed appropriately and rigorously? 

Reviewer #1: Yes

Reviewer #2: Yes

4. Have the authors made all data underlying the findings in their manuscript fully available?

Reviewer #1: Yes

Reviewer #2: Yes

5. Is the manuscript presented in an intelligible fashion and written in standard English?

Reviewer #1: Yes

Reviewer #2: Yes

6. Review Comments to the Author

Reviewer #1: (No Response)

Reviewer #2: The authors have written a satisfying response to my comments. I have no further comments on this revised manuscript.

7. PLOS authors have the option to publish the peer review history of their article (what does this mean?). If published, this will include your full peer review and any attached files.

Reviewer #1: **Yes: **Annalisa Perna

Reviewer #2: No

---

## [Editor Report · Acceptance letter]

16 Sep 2021

PONE-D-21-10078R1 

Somatostatin Analog Therapy Effectiveness on the Progression of Polycystic Kidney and Liver Disease: A Systematic Review and Meta-analysis of Randomized Clinical Trials 

Dear Dr. Suwabe:

I'm pleased to inform you that your manuscript has been deemed suitable for publication in PLOS ONE. Congratulations! Your manuscript is now with our production department. 

Kind regards, 

on behalf of

Prof. Giuseppe Remuzzi 

Academic Editor

PLOS ONE